# Effect of Chronic Ankle Instability on the Biomechanical Organization of Gait Initiation: A Systematic Review

**DOI:** 10.3390/brainsci13111596

**Published:** 2023-11-17

**Authors:** Mohammad Yousefi, Shaghayegh Zivari, Eric Yiou, Teddy Caderby

**Affiliations:** 1Faculty of Sport Sciences, University of Birjand, Birjand 9717434765, Iran; m.yousefi@birjand.ac.ir (M.Y.); shaghayegh.zivari@birjand.ac.ir (S.Z.); 2Complexité, Innovation, Activités Motrices et Sportives (CIAMS), Université Paris-Saclay, 91400 Orsay, France; 3Complexité, Innovation, Activités Motrices et Sportives (CIAMS), Université d’Orléans, 45067 Orléans, France; 4Laboratoire IRISSE—EA 4075, UFR des Sciences de l’Homme et de l’Environnement, Université de La Réunion, 97430 Le Tampon, La Réunion, France; teddy.caderby@univ-reunion.fr

**Keywords:** ankle injury, anticipatory postural adjustments, muscle activity, center of pressure, gait, locomotion

## Abstract

This systematic review was conducted to provide an overview of the effects of chronic ankle instability (CAI) on the biomechanical organization of gait initiation. Gait initiation is a classical model used in the literature to investigate postural control in healthy and pathological individuals. PubMed, ScienceDirect, Scopus, Web of Science, and Google Scholar were searched for relevant articles. Eligible studies were screened and data extracted by two independent reviewers. An evaluation of the quality of the studies was performed using the Downs and Black checklist. A total of 878 articles were found in the initial search, but only six studies met the inclusion criteria. The findings from the literature suggest that CAI affects the characteristics of gait initiation. Specifically, individuals with CAI exhibit notable differences in reaction time, the spatiotemporal parameters of anticipatory postural adjustments (APAs) and step execution, ankle–foot kinematics, and muscle activation compared to healthy controls. In particular, the observed differences in APA patterns associated with gait initiation suggest the presence of supraspinal motor control alterations in individuals with CAI. These findings may provide valuable information for the rehabilitation of these patients. However, the limited evidence available calls for caution in interpreting the results and underscores the need for further research.

## 1. Introduction

Lateral ankle sprains (LASs) are one of the most common musculoskeletal injuries in athletes and the general population [1,2]. Although the majority of people recover from their first LAS (often referred to as “copers” [3,4,5,6]), it has been reported that up to 40% of injured individuals develop chronic ankle instability (CAI) [7,8]. CAI is characterized by recurrent ankle sprains and a subjective feeling of the ankle “giving way” and also by residual symptoms (pain, weakness, and an altered range of motion in the ankle) that persist for at least one year following the initial injury [8,9]. CAI leads to numerous negative health consequences, including a decrease in the quality of life, reduced physical activity levels, and an increased risk of developing posttraumatic ankle osteoarthritis [9,10]. The prevalence and associated consequences make CAI a major public health issue.

Several pieces of evidence suggest that CAI originates from mechanical and sensorimotor impairments [8,9]. Mechanical impairments include pathologic joint laxity due to the loss of the ligamentous complex function and also arthrokinematic and osteokinematic restrictions [11,12]. Sensorimotor alterations (also called functional alterations) are characterized by diminished somatosensation, the presence of pain, altered reflexes, and muscle weakness [8]. Although both the mechanical and sensorimotor alterations may be linked, it is important to note that they can exist independently of each other [13]. Furthermore, it has been shown these CAI-related impairments lead to altered balance control and altered movement patterns in a high number of functional tasks, including walking, landing, and cutting [14,15,16,17,18]. As a result, CAI affects the ability to perform daily activities, and can also lead to an increased risk of falls [19]. Investigating the alterations elicited by CAI in the performance of these daily activities can be helpful in designing interventions aimed at restoring normal movement patterns and reducing the risk of further injuries.

Gait initiation, which represents the transition from a quiet stance to steady-state walking, is a functional task of daily living [20,21,22]. It is also a classical model used in the literature to investigate postural control in healthy and pathological individuals [23]. This task can be divided into two distinct phases: a postural phase preceding the heel-off of the swing leg, corresponding to the so-called anticipatory postural adjustments (APAs), followed by a step execution phase ending at the time of swing foot contact [24,25,26]. The APAs are centrally initiated dynamic phenomena occurring before the onset of the intentional movement [20,25]. During gait initiation, these APAs are manifested as a shift of the center of pressure (COP) backward and laterally toward the swing foot, acting to propel the center of mass (COM) forward and toward the stance-leg side [25,26,27]. It is generally admitted that the goals of these anticipatory dynamic phenomena are twofold: (i) generate the initial propulsive forces necessary to reach the intended velocity progression (or “motor performance”) and (ii) promote postural balance during the subsequent step execution [23,28,29,30,31]. Postural balance is indeed particularly challenged during step execution due to the natural tendency of the COM to fall laterally toward the swing leg side under gravity’s effect. A modeling study recently showed that APA acts to attenuate this lateral fall by shifting the COM nearly above the stance foot at the time of step execution [32].

The APAs are sub-served by a motor synergy characterized by a complex sequence of muscle activation/deactivation in the lower limbs. Ankle muscles play a major role in this motor synergy. Consequently, any alteration of muscle activity in these joints may have negative consequences on motor performance and/or postural balance. In healthy subjects, the anticipatory backward COP shift has been ascribed to the bilateral deactivation of the soleus, followed by the strong activation of both tibialis anteriors [33]. The anticipatory lateral COP shift has been classically attributed to the loading of the swing leg associated with the activation of swing hip adductors [23]. Recent studies reported that the slight flexion of the stance knee and hip during APA also contributes to this action. The flexion of the stance knee is favored by bilateral soleus silencing and greater ipsilateral tibialis anterior activity with respect to contralateral activity, while stance hip flexion is associated with activation of the stance rectus femoris [31].

Quantitative analysis of APAs and muscular activity during gait initiation can provide important insights into the supraspinal motor control mechanisms [22]. Previous research has identified altered neuromuscular strategies in individuals with CAI, including changes in muscle activation patterns during walking, landing, and cutting tasks [34,35,36]. Specifically, some studies have observed decreased muscle activity in the tibialis anterior, medial gastrocnemius, and fibularis longus [36], while others have reported increased muscle activation in these muscles [35]. Although some reviews have examined the influence of CAI on the biomechanics of walking [37,38] or other functional tasks [39], to the best of our knowledge, no study has systematically reviewed the effects of CAI on the biomechanical organization of gait initiation.

The aim of this article was thus to provide an up-to-date literature review of studies focusing on this research question. Understanding how CAI affects gait initiation may contribute to a better understanding of the central mechanisms underlying the development of CAI and may provide clinicians valuable information for the development of targeted rehabilitation therapies [40,41].

## 2. Materials and Methods

This review was conducted following the Preferred Reporting Items for Systematic Reviews and Meta-Analyses (PRISMA) recommendations [42]. The study protocol was registered on the Open Science Framework on 22 October 2023 (https://doi.org/10.17605/OSF.IO/7PTKY).

### 2.1. Search Strategy

An electronic database search was performed by the primary investigator (S.Z.) between 1 June and 18 July 2022 without publication status or publication date restrictions. Five scientific databases were searched: PubMed, ScienceDirect, Scopus, Web of Science, and Google Scholar. Only articles written in English were reviewed. The search terms included: (“chronic ankle instability” OR “ankle instability”) AND (“initiation of gait” OR “gait initiation”). Articles identified from the search were stored and managed using EndNote X8 throughout the review process.

### 2.2. Eligibility Criteria

#### 2.2.1. Inclusion Criteria 

Only peer-reviewed articles meeting the following criteria were included: (1) studies including a group of participants who were diagnosed with CAI (functional or mechanical instability); (2) studies comparing participants with CAI with healthy controls, i.e., without a history of ankle sprain; and (3) studies including either muscle activity, kinematics, or kinetics during gait initiation as main outcome measures.

#### 2.2.2. Exclusion Criteria 

Articles were excluded if: (1) participants had any disorder or pathology other than CAI; (2) studies involved a treatment protocol without preintervention comparison between the CAI and the control groups; (3) studies did not investigate gait initiation; or (4) studies were case studies, case reports, conference papers, or book chapters.

### 2.3. Selection Process

After duplicates were manually removed from the EndNote library, all articles were independently screened by two independent reviewers (M.Y. and S.Z.) according to the eligibility criteria. The selection was conducted first considering the titles and abstracts of the articles. Then, the full texts were checked to examine whether the articles met the inclusion criteria. Reference lists were manually checked to identify additional relevant articles. Any disagreement during the selection process was resolved by a consensus or, if necessary, an additional examiner.

### 2.4. Data Collection Process and Data Extraction

Two reviewers (M.Y. and S.Z.) independently extracted the data. The data extracted included study design, population (sample size and demographic information of participants like gender and age), experimental protocol, outcome measures, and key findings. The extracted data from each study are reported in Table 1.

### 2.5. Risk of Bias and Methodological Quality Assessment

The risk of bias and methodological quality of the included studies were independently assessed by two reviewers (M.Y. and S.Z.) using the Downs and Black Checklist. The Downs and Black Checklist was developed to evaluate the risk of bias for non-randomized and randomized control trials. This checklist consists of 27 items, including reporting, external validity, bias, confounding, and power. Most questions were rated either as “yes” (=1) or “unable to determine/no” (=0), except for item five, which was rated on a 3-point scale (yes = 2, partial = 1, and no = 0). The percentage score was calculated as the ratio between the achieved score and the maximum possible score, multiplied by 100%. Based on the percentage of items met, each study was classified by quality: low (<60%), moderate (60–74%), or high (≥75%). 

## 3. Results

### 3.1. Study Selection and Characteristics

A total of 878 articles were found in the initial search and 717 remained after excluding duplicates. Following title and abstract screening, 21 articles were submitted to a full-text assessment. Two additional studies were added based on reference lists, and after that, 18 studies remained. These studies were checked based on the exclusion criteria, and ultimately, six studies met the inclusion criteria. Figure 1 shows a flow diagram of the study selection process.

A total of 124 patients with unilateral CAI (72 women, 32 men, 20 unspecified, aged 18.5 to 23.5), 126 healthy participants (72 women, 34 men, 20 unspecified, aged 19 to 24), 17 people with a history of a LAS (nine women, eight men, aged 19 to 23), and 21 coper patients (13 women, 8 men, aged 18 to 22) were included. In this systematic review, only comparative data between CAI and healthy participants were considered and are summarized in Table 1. 

### 3.2. Quality Assessment

Kappa coefficients were calculated to determine the inter-rater reliability of the two investigators. The overall agreement between the raters was excellent (κ = 0.924). Each study’s score on the modified Downs and Black checklist is presented in Table 2. The quality of the studies based on the average Checklist was 11, with a range of 9–13. No study was rated as having excellent methodological quality. Most studies required a history of ankle sprain at least one year prior to testing and a score of <90% in activities of daily living and of <80% in sports activities from the foot and ankle ability measure (FAAM) questionnaire as inclusion criteria for the CAI group. The absence of a history of lower limb extremity or neurological disorders was an inclusion criterion for the healthy group, although this was not specified in one study [26].

### 3.3. Results of Studies

Among the selected studies, three focused on gait initiation triggered in response to an auditory signal [26,30,41], two studies examined both self-generated and externally triggered gait initiation [22,29], and one provided no information on the mode of gait initiation (i.e., self-generated or triggered by an external stimulus) [21]. All studies examined gait initiation in the forward direction, and two of them also examined gait initiation in two additional directions (30° in the medial direction and 30° in the lateral direction) [26,29]. Gait was initiated at a self-selected speed in three studies [21,30,41] and at maximum speed in the other three studies [22,26,29]. Gait was initiated with the injured limb in two studies [21,41], with the non-injured limb in one study [29], and with both limbs in three studies [22,26,30].

*Reaction time.* Two of the included studies investigated the reaction time, i.e., the delay between the trigger signal and the onset of APA, during forward gait initiation with the injured leg at a spontaneous speed [41] and during multi-directional gait initiation with the non-injured limb at a maximal speed [29]. These two studies revealed that participants with CAI had a longer reaction time than healthy controls.

*APA temporospatial features.* Among the selected studies, two examined the temporal aspects of APA [29,41]. These studies reported that individuals with CAI exhibited a shorter APA duration compared to those without CAI during externally triggered forward gait initiation with the injured leg at a spontaneous speed [41] and during gait initiation with the non-injured limb at a maximal speed in various directions and triggering conditions [29].

Five studies have investigated the kinematics of the COP in individuals with CAI [22,26,29,30,41]. Among these studies, two found a significant decrease in the peak lateral COP shift in these individuals as compared to healthy controls during gait initiation at maximal speed, without any influence of the involved limb (injured and non-injured) [22,26], mode of triggering (self-selected or externally triggered) [22], or the direction of gait initiation (medial, forward, or lateral) [26]. This decrease in the lateral COP displacement was more predominant when gait was initiated with the injured leg than with the non-injured leg in CAI [22,26]. Another study also reported a decrease in the posterolateral COP displacement in the APA phase in people with CAI during externally triggered forward gait initiation at a spontaneous speed, but only when gait was initiated with the non-injured leg and not the injured leg [30]. In contrast, of these five studies, two found no significant difference in the lateral COP shift during externally triggered forward gait initiation with the injured leg at spontaneous speed [41] and during gait initiation with the non-injured limb at a maximal speed in various directions and triggering conditions [29]. In addition, all five of these studies reported no difference in the anteroposterior COP shift during the APA phase between CAI and healthy participants. Of the included studies, two examined the kinematics of the COM in the APA phase and found no significant difference in the anteroposterior COM velocity at the end of APA between CAI patients and healthy people during gait initiation at a maximum speed, regardless of the involved limb [26], mode of triggering [29], or the direction of gait initiation [26,29]. One study reported a smaller vertical COM velocity during APA in CAI patients compared to healthy controls during externally triggered gait initiation at maximum speed in the forward and medial directions, regardless of the involved limb [26].

*Muscle activation.* Only one study investigated the effect of CAI on muscle activity during gait initiation [41]. This study found an earlier activation of the soleus muscle of the injured limb after the APA onset in CAI participants as compared to healthy participants during externally triggered forward gait initiation with the injured limb at a self-selected speed, suggesting a decreased inhibition duration of this muscle in people with CAI [41]. 

*Spatiotemporal parameters of step execution*. Two studies have investigated the characteristics of the execution phase of gait initiation in individuals with CAI [22,30]. One study found a decrease in the mediolateral COP displacement toward the stance foot during the swing-foot lift phase (from heel-off to toe-off) when gait was initiated forward at a self-selected speed with the non-injured leg (i.e., when the injured leg served as the stance leg) [30]. Moreover, the same study found that the mediolateral displacement and velocity of COP during the unipedal phase of the step execution were greater when gait was initiated with the non-injured leg than with the injured leg in individuals with CAI [30]. Another study reported that the maximal forward COP velocity during the execution phase was greater in CAI patients than in healthy controls when gait was initiated at maximal speed with the non-injured leg in both self-generated and externally triggered gait conditions [22].

*Ankle–foot kinematics.* One article examined the multisegmented ankle–foot kinematics during forward gait initiation at a self-selected speed in CAI, LAS, and control participants [21]. The results showed the CAI group had increased rearfoot inversion from 34% to 91% of the stance phase, i.e., from the heel strike to the foot-off of the injured leg (taking the first step), compared to healthy controls. 

## 4. Discussion

The aim of this article was to provide an up-to-date literature review of studies focusing on the effects of CAI on the biomechanical organization of gait initiation. Overall, this review suggests that CAI affects characteristics of gait initiation, including reaction time, the spatio-temporal parameters of APA and step execution, ankle–foot kinematics, and muscle activation.

The study of APA has been proposed as a means of gaining valuable insights into the supraspinal motor control mechanisms [22,26,41], as these anticipatory postural phenomena appear to be mediated centrally by supraspinal centers [43,44]. The present systematic review suggests that the spatiotemporal features of APA are affected by CAI. Specifically, the findings from the literature reveal a decrease in the APA duration in CAI patients compared to healthy individuals [29,41]. Regarding the APA amplitude, the majority of the studies reported a reduction in the magnitude of COP displacement during APA, in particular in the mediolateral direction, in CAI patients as compared to healthy controls [22,26,30]. This decrease in both the duration and mediolateral amplitude of APA in CAI people has been reported across multiple gait initiation directions and in self-generated and externally triggered conditions [26,29,41]. Furthermore, these CAI-related changes in the APA are independent of the limb initiating gait (injured or uninjured limb), although the effects appear to be predominant when gait is initiated with the injured leg (i.e., the injured leg taking the first step) [22,26]. These results evidence bilateral alterations in the APA in people with unilateral CAI and support the hypothesis of impaired supraspinal motor control mechanisms in these patients. Despite the changes observed in the APA amplitude along the mediolateral direction, the literature unanimously reported that CAI, on the other hand, caused no change in the anteroposterior COP displacement during APA [22,26,29,30,41]. It is well known that APAs in this direction generate an initial COM velocity that predetermines the forward progression velocity at the end of the first step [45,46]. Consistent with the absence of change in the anteroposterior APA with CAI, studies found no difference in the forward COM velocity at the end of APA between CAI and healthy individuals [26,29]. Interestingly, it is presumed that the APAs along the anteroposterior direction are implemented by ankle synergy, characterized by a bilateral inhibition of the soleus followed by a strong bilateral activation of the tibialis anterior [33]. In the literature, only one study examined the effects of CAI on muscle activation during gait initiation [41]. This study showed that CAI patients had an earlier activation of the soleus muscle of the injured limb (taking the first step) during the APA phase as compared to healthy participants, indicating a decrease in the soleus inhibition duration in these patients. This observation could be related to the reduced APA duration in patients with CAI. Nevertheless, despite this shorter APA duration, suggesting a shorter duration to propel the COM forward, CAI patients seem to be able to produce a forward COM velocity at the end of APA similar to that of healthy participants [26,29]. Although no study has quantified the forward progression velocity at the end of the first step (often referred to as motor performance), these findings suggest that CAI does not affect the ability of APA to generate convenient conditions for forward progression during gait initiation.

The reason why CAI affects only the mediolateral component of APA and not the anteroposterior dimension remains unclear. Previously, it has been shown that the lateral COP shift during the APA phase of gait initiation is mainly ascribed to the activation of the hip adductors/abductors and the knee flexion of the initial stance leg [31,47,48]. Nevertheless, it cannot be ruled out that the invertor/evertor muscles of the ankle also contribute to the lateral COP motion during APAs, as has already been observed during walking and standing [49,50]. Interestingly, it has been shown that CAI is associated with weakness and activation abnormalities of these muscles [39,51], which play an important role in ankle stability. Thus, it may be hypothesized that the alterations observed in people with CAI in mediolateral APAs could be associated with strength and recruitment deficits in these ankle invertor/evertor muscles. Furthermore, as some authors speculate [30], it is likely that these APA alterations are also linked to a compensatory strategy adopted by CAI patients to minimize anticipatory postural forces. These adaptations could then be seen as a safety strategy implemented by these patients to reduce the risk of ankle sprains [52]. Although the underlying mechanisms of APA alterations in CAI patients remain to be elucidated, it should be borne in mind that alterations in the mediolateral APA may potentially affect postural stability during step execution.

The step execution phase of gait initiation is often assimilated to a ballistic phase, during which the body falls forward and toward the swing leg under the action of gravity, pivoting around the ankle joint [31,32,40,46]. It is generally admitted that the lateral COP shift during APA acts to propel the COM near the stance foot before the foot-off of the swing leg, thus reducing the lateral fall of the body, i.e., lateral instability, during the subsequent step execution. In the absence of this anticipatory postural dynamic, lateral instability would be greater, which could potentially increase the risk of falling during gait initiation [32,53,54]. In the literature, we found no study quantifying the mediolateral stability during step execution in CAI patients. However, several studies compared the COP motion during this phase between CAI and healthy participants [22,30]. A decrease in the lateral COP displacement toward the stance foot during the unloading phase of the execution phase (often referred as to the foot-lift phase [27,55]) has been reported in CAI patients compared to healthy people [30]. This result can probably be explained in part by the fact that, as the lateral COP displacement toward the swing foot during APA is less important in CAI patients, the lateral distance that the COP must then travel to position itself under or close to the stance foot before the swing foot-off is also less important. Nevertheless, it has been revealed that CAI patients had a faster forward COP velocity during the unipedal phase than healthy individuals when gait was initiated with the non-injured leg, i.e., when the injured leg served as the stance leg [22]. In addition, other authors showed that the lateral COP velocity during this phase was faster when gait was initiated with the non-injured leg than with the injured leg [30]. Although none of these studies quantified the duration of the step execution phase, it can be hypothesized that these adaptations in CAI patients may serve to reduce the duration of step execution and thus reduce postural demands during this phase [22,30]. The presence of supraspinal adaptations in individuals with CAI suggests that it should be treated as a global condition (rather than solely as a local musculoskeletal condition) affecting all levels of the neuromuscular control system [30,41]. Rehabilitation programs that improve neuromuscular control and restore normal muscle activation patterns may reduce the risk of ankle sprains and other injuries in individuals with CAI [30]. Interestingly, recent studies have reported that APA-focused training is effective in enhancing APA [56,57]. This type of training could be used to address supraspinal aspects of motor control in the management of CAI. In particular, it could be an attractive approach to improving balance and movement control in patients with CAI. However, this remains to be verified.

Our study has certain limitations that must be acknowledged. The number of studies included in this review is relatively small (n = 6), these studies were of moderate quality, and the conditions tested in these articles are quite different from one another (e.g., self-selected and triggered conditions, speed conditions, etc.). However, in general, the majority of the studies found that characteristics of gait initiation were affected by CAI. It is important that future studies consider quantifying postural stability using classical biomechanical variables (e.g., margin of stability [27], braking index [55], etc.) and controlling for gait initiation speed and the mode of triggering, as these factors can be confounding [27,28,29,30,31,32,33,34,35,36,37,38,39,40,41,42,43,44,45,46,47,48,49,50,51,52,53,54,55,56,57,58]. Furthermore, it should be borne in mind that all studies included in this review were retrospective in nature, making it difficult to determine whether the changes in the characteristics of gait initiation found in patients with CAI existed prior to the initial ankle sprain or if they were the result of the injury.

## 5. Conclusions

The present systematic review suggests that CAI affects the characteristics of gait initiation, including reaction time, the spatiotemporal parameters of APAs and step execution, ankle–foot kinematics, and muscle activation. These findings may provide valuable information for the rehabilitation of these patients. However, the limited evidence available calls for caution in interpreting the results and underscores the need for further research.

## Figures and Tables

**Figure 1 brainsci-13-01596-f001:**
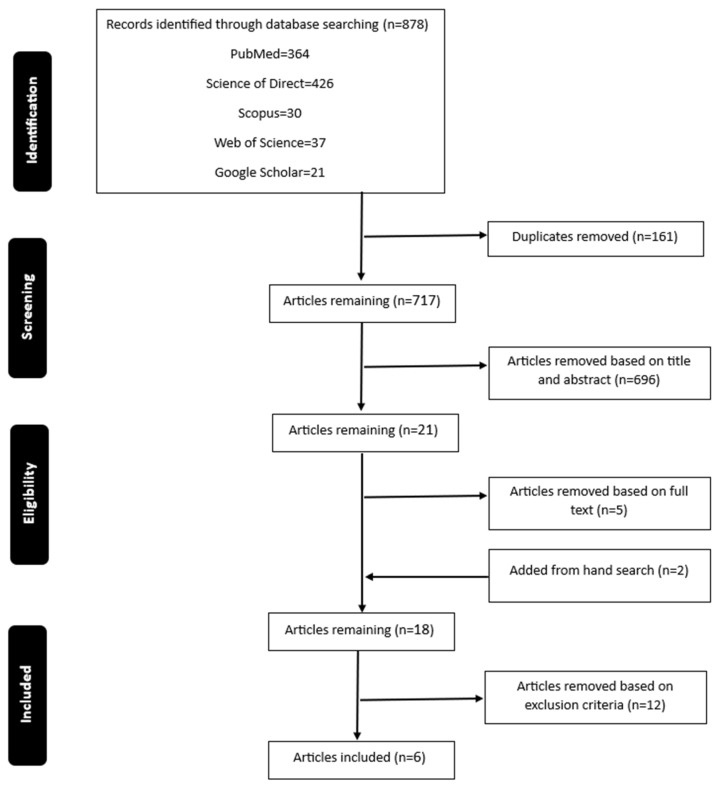
Flow chart of the selection process.

**Table 1 brainsci-13-01596-t001:** Characteristics of the included studies.

Study	Study Design	Participants	Protocol	Outcome Measures	Key Findings
Ebrahimabadi et al., 2017 [22]	Cross-sectional study	22 CAI (F:22, 22.4 ± 1.5 yrs) 22 healthy (F:22, 22.7 ± 1.8 yrs)	Triggered GI at maximum speed with both the injured and non-injured limb.	Displacement and velocity of the COP during APA and execution phases.	Peak ML COP displacement toward the swing leg in the APA phase of GI was reduced in CAI. Forward COP velocity was increased in CAI in the execution phase of GI.
Ebrahimabadi et al., 2018 [26]	Pilot cross-sectional study	20 CAI (21.4 ± 1.3 yrs) 20 healthy (21.7 ± 1.5 yrs)	Triggered GI at maximum speed in 3 directions (forward, 30° medial, and 30° lateral) with both the injured and non-injured limb.	COP and COM kinematics.	AP COM velocity at the end of APA did not differ between CAI and controls. Peak ML COP shift and vertical COM velocity during APA were decreased in CAI.
Ebrahimabadi et al., 2022 [29]	Cross-sectional study	25 CAI (F:20/M:5, 22.01 ± 1.08 yrs) 25 healthy (F:21/M:4, 22.90 ± 1.61 yrs)	Triggered and self-generated GI at maximum speed in 3 directions (forward, 30° medial, and 30° lateral) with the non-injured limb.	Reaction time and APA phase durations, COP displacement, and COM velocity during the APA phase.	Longer reaction time and shorter APA duration (7%) in CAI. No difference in COP displacement and COM velocity between CAI and controls.
Fraser et al., 2019 [21]	Cross-sectional study	22 Control (F:13/M:9, 19.6 ± 0.9 yrs) 17 LAS (F:9/M:8, 21.0 ± 2.3 yrs) 21 Coper (F:13/M:8, 20.8 ± 2.9 yrs) 20 CAI (F:15/M:5, 19.8 ± 1.3 yrs)	GI at a self-selected speed with the injured limb.	Three-dimensional kinematics of the hallux, forefoot, midfoot, and rearfoot.	Rearfoot inversion during the end of step execution phase increased by 5.3° in CAI.
Hass et al., 2010 [30]	Cross-sectional study	20 CAI (F:15/M:5, 20.5 ± 61.0 yrs) 20 Control (F:16, M:4, 20.85 ± 61.6 yrs)	Triggered GI at a self-selected speed with both the injured and non-injured limb.	Displacement and velocity of the COP during the APA and execution phases.	Resultant COP displacement in the APA phase and ML COP displacement in the execution phase were reduced in CAI when gait was initiated with the non-injured limb.
Yousefi et al., 2020 [41]	Cross-sectional study	17 CAI (M:17, 24.31 ± 0.81 yrs) 17 Control (M:17, 23.40 ± 1.70 yrs)	Triggered gait initiation at a self-selected speed with the injured limb.	Reaction time and APA duration, COP excursion, muscle activation.	Longer reaction time phase and shorter APA duration in CAI. No difference in AP and ML normalized peak COP excursions in the APA phase. Earlier soleus activation in the injured limb in CAI.

LAS = lateral ankle sprain; GI = gait initiation; APA = anticipatory postural adjustments; ML = mediolateral; AP = anteroposterior; COP = center of pressure; COM = center of mass.

**Table 2 brainsci-13-01596-t002:** Modified Downs and Black quality index results and total score.

	Quality items	Ebrahimabadi et al., 2022 [29]	Yousefi et al., 2020 [41]	Fraser et al., 2019 [21]	Ebrahimabadi et al., 2018 [26]	Ebrahimabadi et al., 2017 [22]	Hass et al., 2010 [30]
Reporting	Q1	1	1	1	1	1	1
	Q2	1	1	1	1	1	1
	Q3	1	1	1	1	1	1
	Q4	0	0	0	0	0	0
	Q5	2	2	2	2	2	2
	Q6	0	1	1	1	0	1
	Q7	1	1	1	1	1	1
	Q8	0	0	0	0	0	0
	Q9	0	0	0	0	0	0
	Q10	1	1	1	1	1	1
External Validity	Q11	1	0	0	0	1	0
	Q12	0	0	0	0	0	0
	Q13	1	0	1	0	1	0
Internal Validity–Bias	Q14	0	0	0	0	0	0
	Q15	0	0	0	0	0	0
	Q16	0	0	0	0	0	0
	Q17	0	0	0	0	0	0
	Q18	1	1	1	1	1	1
	Q19	0	0	1	0	0	0
	Q20	0	0	1	0	0	0
Internal Validity–Confounding	Q21	1	0	1	0	0	1
	Q22	0	0	0	0	1	0
	Q23	0	0	0	0	1	0
	Q24	0	0	0	0	0	0
	Q25	0	0	0	0	0	0
	Q26	0	0	0	0	0	0
Power	Q27	0	0	0	1	1	0
Total		11	9	13	10	13	10

Quality items = Q1: Hypothesis/Aim. Q2: Main Outcomes in Method/Introduction. Q3: Inclusion/Exclusion Criteria. Q4: Description of Interventions. Q5: Description of principal confounders. Q6: Main Findings. Q7: Random Variability. Q8: adverse events. Q9: Lost to Follow-Up. Q10: Actual Probability Values. Q11: Representative of the Entire Population (subjects asked to participate in the study). Q12: Representative of the Entire Population (subjects who were prepared to participate) Q13: Representative of the treatment (staff, places, and facilities). Q14: Blind Study Subjects. Q15: blind those measuring. Q16: data dredging Q17: analyses adjust for different lengths of follow-up of patients. Q18: Statistical Tests Appropriated Q19: reliable compliance with the intervention. Q20: Outcome Measures Used Accurate Q21: patients in different intervention groups or from the same population? Q22: same period of time Q23: Random Allocation Q24: assignment concealed Q25: adequate adjustment for confounding Q26: Losses of Patients to Follow-up. Q27: Estimate of Statistical Power.

## Data Availability

Not applicable.

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
