# Peer review of "Effect of Chronic Ankle Instability on the Biomechanical Organization of Gait Initiation: A Systematic Review"

_brainsci, 2023, doi:10.3390/brainsci13111596_

Round 1
Reviewer 1 Report
Comments and Suggestions for Authors
The article entitled “Effect of chronic ankle instability on the biomechanical organization of gait initiation: A systematic review” is interesting and focuses on the effects of chronic ankle instability (CAI) on the biomechanical organization of gait initiation. However, it is difficult to follow and suffers from a series of technical flaws.
I do not think the following sentence “CAI has a significant impact on the characteristics of gait initiation” is totally true, especially when the authors found just 6 articles and none of them have good quality in the experimental methodology. The quality assessment of the manuscripts shows that the articles do not comply with most of the quality items. The authors must decide if the reported outcomes have a significant contribution to the scientific community. The results cannot be generalized due to the small number of manuscripts analyzed. Furthermore, it is not clear how the findings of the review provide valuable information for the rehabilitation of the patients. Lateral ankle sprains and chronic ankle stability are two different scenarios, they can be related up to 40% but are not the same. This may have an impact on the search strategy and could give a different number of manuscripts found. It seems that the authors are missing the other 60% of the information. Then, the results and discussion sections will change.
In the materials and methods section, lines 101-102, add the reference for the PRISMA 2020 recommendations.
Please follow the PRISMA 2020 guidelines for systematic reviews. https://doi.org/10.1136/bmj.n71 “Systematic reviews serve many critical roles. They can provide syntheses of the state of knowledge in a field, from which future research priorities can be identified; they can address questions that otherwise could not be answered by individual studies; they can identify problems in primary research that should be rectified in future studies; and they can generate or evaluate theories about how or why phenomena occur. Systematic reviews therefore generate various types of knowledge for different users of reviews (such as patients, healthcare providers, researchers, and policy makers).”
It is not clear how the inclusion criteria were defined, which type of instabilities were considered to include the articles, and whether studies with just ankle sprains were considered.
In the Risk of Bias and Methodological Quality Assessment, line 8, indicates the score for unable to determine.
Why the search strategy did not consider the phrase “Anticipatory Postural Adjustments”?
In the results section, line 16. Why the two articles mentioned by the authors were included by reading the references, why those two articles were not found during the search strategy?
Revise the number of pages and line numbers.
Results section, lines 21-22, revise the number of healthy participants. The authors have to write the results giving the importance of the findings and explaining the effect of CAI and LAS in gait initiation, not just rewriting the results found by other authors.
The conclusions cannot be supported by the outcomes found in the review as the authors do not know if the changes in the characteristics of gait initiation found in patients with CAI existed prior to the initial ankle sprain or if they were the result of the injury. Furthermore, the question set at the beginning of the study is not completely answered as they do not mention rehabilitation protocols that can help to recover the CAI.
Comments on the Quality of English LanguageMinor editing of English language required
Author Response
Response to Reviewer #1
We thank the reviewer for his/her thoughtful review of our manuscript and his relevant comments. Please find below the point-by-point response to comments.
The article entitled “Effect of chronic ankle instability on the biomechanical organization of gait initiation: A systematic review” is interesting and focuses on the effects of chronic ankle instability (CAI) on the biomechanical organization of gait initiation. However, it is difficult to follow and suffers from a series of technical flaws.
I do not think the following sentence “CAI has a significant impact on the characteristics of gait initiation” is totally true, especially when the authors found just 6 articles and none of them have good quality in the experimental methodology. The quality assessment of the manuscripts shows that the articles do not comply with most of the quality items. The authors must decide if the reported outcomes have a significant contribution to the scientific community. The results cannot be generalized due to the small number of manuscripts analyzed. Furthermore, it is not clear how the findings of the review provide valuable information for the rehabilitation of the patients. Lateral ankle sprains and chronic ankle stability are two different scenarios, they can be related up to 40% but are not the same. This may have an impact on the search strategy and could give a different number of manuscripts found. It seems that the authors are missing the other 60% of the information. Then, the results and discussion sections will change.
RESPONSE: We acknowledge that the results of the present review should be interpreted with caution, due to the small number, moderate quality and substantial heterogeneity of the included studies. This point is now highlighted in our revised manuscript. Please see abstract (lines 24-26) and conclusion (page 12, lines 373-777).
In addition, based on the reviewer’s remark, we have also clarified the aspects relating to the application of the results for patient rehabilitation. We emphasize that the results obtained may encourage the proposal of interventions targeting the Anticipatory Postural Adjustments (APA) in patients with CAI. In particular, APA enhancement may be a relevant approach to improving balance and movement control in these patients. Protocols have previously been investigated in the literature and could be tested in patients with CAI. Please see page 11, lines 354-359.
Finally, following the remark of the reviewer, we realized that the target population for this review was not entirely clear. We now clarify that this review focused specifically on CAI (and not on lateral ankle sprains). To clarify this point, we have detailed the inclusion criteria in our review (please see page 3, lines 113-114). In addition, we now mention that only comparative data between CAI and healthy participants (and not LAS and copers) are included in our systematic review. Please see page 6, lines 161-162.
We thank the reviewer for these relevant remarks that help us to improve the quality of our work.
In the materials and methods section, lines 101-102, add the reference for the PRISMA 2020 recommendations. Please follow the PRISMA 2020 guidelines for systematic reviews. https://doi.org/10.1136/bmj.n71 “Systematic reviews serve many critical roles. They can provide syntheses of the state of knowledge in a field, from which future research priorities can be identified; they can address questions that otherwise could not be answered by individual studies; they can identify problems in primary research that should be rectified in future studies; and they can generate or evaluate theories about how or why phenomena occur. Systematic reviews therefore generate various types of knowledge for different users of reviews (such as patients, healthcare providers, researchers, and policy makers).”
RESPONSE: Thank you for this suggestion. This reference has been added to the revised manuscript. Please see page 3, line 100.
It is not clear how the inclusion criteria were defined, which type of instabilities were considered to include the articles, and whether studies with just ankle sprains were considered.
RESPONSE: As mentioned above, this review focused specifically on chronic ankle instability (and not on ankle sprains). Consequently, the studies selected had to include a group of participants diagnosed with CAI (mechanical or functional instability). Following the remark of the reviewer, we now specify this point in the inclusion criteria in the revised manuscript (please see page 3, lines 113-114). In addition, our now specify than only comparative data between CAI and healthy participants (and not LAS and copers) are considered in our systematic review. Please see page 6, lines 161-162.
In the Risk of Bias and Methodological Quality Assessment, line 8, indicates the score for unable to determine.
RESPONSE: Thank you for this suggestion. This score has been added. Please see page 6, line 145.
Why the search strategy did not consider the phrase “Anticipatory Postural Adjustments”?
RESPONSE: Thank you for this relevant remark. The objective of our study was to systematically review studies investigating the effects of CAI on the biomechanics of gait initiation, including the kinematics and kinetics aspects of Anticipatory Postural Adjustments and step execution. Therefore, we did not focus specifically on Anticipatory Postural Adjustments. Please note that the majority of studies included in the present review (4 out of 6) did not use the terms “Anticipatory Postural Adjustments” in their title, abstract and keywords. Adding these terms in our seach strategy (in addition to gait initiation) would not have increased the number of “relevant” results. To check this, we launched a Pubmed search with the addition of these terms to our query. This search revealed that adding the terms “Anticipatory Postural Adjustments” to the query did not change the number of "relevant" results (4 of the 6 studies included in our review, the other two being found in another database), but only yielded additional articles that did not address gait initiation (for example, leg abduction, standing on roller skates and kicking ball tasks).
In the results section, line 16. Why the two articles entioned by the authors were included by reading the references, why those two articles were not found during the search strategy?
RESPONSE: These articles were not found in our initial search, as they were not referenced in the databases we queried.
For your information, although these two articles were added by cross-referencing, neither was included in the review.
Revise the number of pages and line numbers.
RESPONSE: Thank you for this remark. The number of page and line was revised throughout the revised manuscript.
Results section, lines 21-22, revise the number of healthy participants. The authors have to write the results giving the importance of the findings and explaining the effect of CAI and LAS in gait initiation, not just rewriting the results found by other authors.
RESPONSE: Thank you for this remark. The number of healthy participants was revised. Please see page 6, line 158.
Furthermore, as recommended by the PRISMA guidelines, we have presented a synthesis of individual study results for each outcome measure in our results section. Please note that, at the editor's request, we have reduced the number of words in the manuscript, including in the results section. In addition, we now present only comparative data between CAI and healthy participants in this section (results concerning LAS were removed).
The conclusions cannot be supported by the outcomes found in the review as the authors do not know if the changes in the characteristics of gait initiation found in patients with CAI existed prior to the initial ankle sprain or if they were the result of the injury. Furthermore, the question set at the beginning of the study is not completely answered as they do not mention rehabilitation protocols that can help to recover the CAI.
RESPONSE: As explained above, we now make it clear that conclusions must be drawn with caution due to the limitations of the present study. Please see page 12, lines 376-377.
In addition, based on the remark of the reviewer, we also clarified the aspects relating to patient rehabilitation. Please see page 11, lines 350-359.
Reviewer 2 Report
Comments and Suggestions for Authors
This systematic review on the "Effect of chronic ankle instability on the biomechanical organization of gait initiation" follow appropriate methodology for a systematic review. To my knowledge there has never been a systematic review on this topic.
The authors followed the PRISMA guidelines and their methodology met the requirements and was fully reported.
The introduction and discussion of this review are comprehensive. Importantly they really try to address mechanism.
It is well written and makes and important contribution to the literature. I especially appreciated the scholarly discussion of the reasons for their overall findings including supraspinal mechanisms. The authors defined their search strategy clearly and they include relevant and appropriate references.
Author Response
Response to Reviewer #2
We thank the reviewer for her/his review of our manuscript and her/his positive attitude toward our work.
This systematic review on the "Effect of chronic ankle instability on the biomechanical organization of gait initiation" follow appropriate methodology for a systematic review. To my knowledge there has never been a systematic review on this topic.
The authors followed the PRISMA guidelines and their methodology met the requirements and was fully reported.
The introduction and discussion of this review are comprehensive. Importantly they really try to address mechanism.
It is well written and makes and important contribution to the literature. I especially appreciated the scholarly discussion of the reasons for their overall findings including supraspinal mechanisms. The authors defined their search strategy clearly and they include relevant and appropriate references.
Reviewer 3 Report
Comments and Suggestions for Authors
Dear authors
Thank you for the opportunity to review the manuscript “
Effect of chronic ankle instability on the biomechanical organization of gait initiation: A systematic review”
It is a great research that has been very nice to me to review it. To improve the quality of the paper, I suggest only a few questions to the authors that are important:
Pag. 12 of material and methods:
“2. Materials and Methods 100
This review was conducted following the Preferred Reporting Items for Systematic 101 Reviews and Meta-Analyses (PRISMA) recommendations.”
Reviewer
It is indicated that the systematics reviews are registered in Prospero or a similar database to be publish, had been registered in anyone? The objective is to promote transparency of methods and reduce bias, which can be reviewed and reduce unnecessary duplication of efforts between researchers.
Pag. 12 of results:
“3. Results 12
3.1. Study selection and characteristics 13
A total of 878 articles were found in initial search and 717 remained after excluding 14 duplicates.”
Reviewer
How were eliminated the duplicates? Was any special programme designed for it used? Please, explain in the text.
Something important to consider is the level of evidence of the topic, I considerated it is relevant to include it. See the document below
“Another thing to consider is the level of evidence of the issue”
The conclusions may be improved respect the level of evidence and shorter.
Best regards
Author Response
Response to Reviewer #3
We thank the reviewer for her/his time and effort in reviewing our manuscript. Please find below a point-by-point response to previous comments.
Dear authors
Thank you for the opportunity to review the manuscript “
Effect of chronic ankle instability on the biomechanical organization of gait initiation: A systematic review”
It is a great research that has been very nice to me to review it. To improve the quality of the paper, I suggest only a few questions to the authors that are important:
Pag. 12 of material and methods:
“2. Materials and Methods 100
This review was conducted following the Preferred Reporting Items for Systematic 101 Reviews and Meta-Analyses (PRISMA) recommendations.”
Reviewer
It is indicated that the systematics reviews are registered in Prospero or a similar database to be publish, had been registered in anyone? The objective is to promote transparency of methods and reduce bias, which can be reviewed and reduce unnecessary duplication of efforts between researchers.
RESPONSE: Thank you for this suggestion. We registered our review on the Open Science Framework. This is now specified in the revised manuscript. Please see page 3, lines 100-101.
Pag. 12 of results:
“3. Results 12
3.1. Study selection and characteristics 13
A total of 878 articles were found in initial search and 717 remained after excluding 14 duplicates.”
Reviewer
How were eliminated the duplicates? Was any special programme designed for it used? Please, explain in the text.
RESPONSE: Duplicates were manually removed from the EndNote Library. This is now specified in the revised manuscript. Please see page 3, line 123.
Something important to consider is the level of evidence of the topic, I considerated it is relevant to include it. See the document below
“Another thing to consider is the level of evidence of the issue”
The conclusions may be improved respect the level of evidence and shorter.
RESPONSE: Thank you for this suggestion. As requested by the reviewer, we have shortened our conclusions and improved them with regard to the level of evidence. Please see page 12, lines 373-377.
We thank the reviewer for these relevant remarks that help us to improve the quality of our work.
Round 2
Reviewer 1 Report
Comments and Suggestions for Authors
Page 6, line 145. Revise the score for "unable to determine", is it cero or two? It cannot be the same score as "no". In Table 1, there is a number two. This produces confusion, then Table 1 has to be modified as well.
Author Response
Response to Reviewer #1
We thank the reviewer for his/her review of our revised manuscript. Please find below the response to his/her comment.
Page 6, line 145. Revise the score for "unable to determine", is it cero or two? It cannot be the same score as "no". In Table 1, there is a number two. This produces confusion, then Table 1 has to be modified as well.
RESPONSE: We thank the reviewer for this remark. We clarified this point in the revised manuscript. Please see page 6, lines 144-146.
Reviewer 3 Report
Comments and Suggestions for Authors
I consider the manuscritp has been improved
Author Response
We thank the reviewer for his/her positive feedback.